# Rheological Properties and Stability of Thickeners for Clinical Use

**DOI:** 10.3390/nu14173455

**Published:** 2022-08-23

**Authors:** Fernando Calmarza-Chueca, Ana Cristina Sánchez-Gimeno, Javier Raso-Pueyo, José Miguel Arbones-Mainar, Alberto Caverni-Muñoz, Alejandro Sanz-Arque, Alejandro Sanz-Paris

**Affiliations:** 1Department of Endocrinology and Nutrition, Miguel Servet Hospital, 50009 Zaragoza, Spain; 2Food Technology, Faculty of Veterinary, AgriFood Institute of Aragon (IA2), Zaragoza University, C/Miguel Servet, 177, 50013 Zaragoza, Spain; 3Aragón Health Research Institute, 50009 Zaragoza, Spain; 4Adipocyte and Fat Biology Laboratory (AdipoFat), Translational Research Unit, Instituto Aragones de Ciencias de la Salud (IACS), University Hospital Miguel Servet, 50009 Zaragoza, Spain; 5Biomedical Research Center in Physiopathology of Obesity and Nutrition (CIBERObn), Institute of Health Carlos III (ISCIII), 28029 Madrid, Spain; 6Diet Service, Renal Patient’s Association, Alcer Ebro, 50009 Zaragoza, Spain

**Keywords:** dysphagia, rheology, thickener, viscosity, predictive model, concentration

## Abstract

The adaptation of liquids for patients with dysphagia requires precision and individualization in the viscosities used. We describe the variations of viscosity in water at different concentrations and evolution over time of the three compositions of commercial thickeners that are on the market (starch, starch with gums, and gum). By increasing the concentration in water, the viscosity of gum-based thickeners increases linearly, but it did not reach pudding texture, whereas the viscosity of the starch-based thickeners (alone or mixed with gums) rapidly reaches very thick textures. We modeled the viscosity at different concentrations of the four thickeners using regression analysis (*R*^2^ > 0.9). We analyzed viscosity changes after 6 h of preparation. The viscosity of gum-based thickeners increased by a maximum of 6.5% after 6 h of preparation, while starch-based thickeners increased by up to 43%. These findings are important for correct handling and prescription. Gum-based thickeners have a predictable linear behavior with the formula we present, reaching nectar and honey-like textures with less quantity of thickener, and are stable over time. In contrast, starch thickeners have an exponential behavior which is difficult to handle, they reach pudding-like viscosity, and are not stable over time.

## 1. Introduction

Dysphagia is a swallowing disease which refers to having problems moving liquids or solids from the mouth to the stomach, and can cause malnutrition, dehydration, choking, and aspiration pneumonia, often resulting in death [1]. One of the main treatments for patients with dysphagia is modifying the viscosity of liquids using commercial thickeners, thereby reducing the risk of aspiration [2]. These thickeners for clinical use can be basically classified into two categories: starch-based thickeners and gum-based thickeners. At first, the thickeners were based exclusively on starch, later thickeners based on starch and a gum-starch mixture were formulated, and finally, thickeners based on gums with maltodextrins appeared [3]. These gum-based thickeners have stability, amylase resistance, texture smoothness, adherence, and safety advantages over starch-based thickeners [4]. Due to diverse compositions with various ingredients, these thickeners present different characteristics in their rheological behavior and in their stability over time, in addition to organoleptic differences [4,5,6,7]. These differences complicate the management of thickeners in daily clinical practice, especially in nursing homes where different types of thickeners coexist. In addition, the preparation of thickeners must be carried out at defined concentrations to achieve the required viscosity. Proper management of the preparation of thickened liquids by caregivers and health professionals is very important to reduce or eliminate the consequences of liquid dysphagia [8]. On many occasions, the preparation of thickened liquids is carried out in an inadequate way, with health personnel tending to make higher viscosities and not checking the viscosity with qualitative methods [9]. Qualitative methods to determine the viscosity reached with a thickener are the most commonly used on a day-to-day basis, but do not provide the accuracy and objectivity of the rheometer [10,11]. Rheology has not become popular in clinical practice because it is a complex technique, requiring technology within the reach of few clinicians [4]. In addition, the latest studies suggest it is important to individualize viscosity patterns as much as possible, and that small changes in viscosity can be detected even by institutionalized elderly patients [12,13]. Knowing the rheological behavior of the thickener and being able to predict the amount needed to reach a given viscosity can help improve the management and treatment of liquid dysphagia. If the viscosity is too high, patients decrease their tolerance to liquids, showing rejection, dehydration, and pharyngeal residues [14]. On the contrary, low viscosity may cause problems with swallowing safety. Therefore, viscosity measured in millipascals should be customized for each patient, which would allow safe and effective swallowing [15].

There are scarce comparative data on the rheological properties of the three different types of commercial thickeners available on the market (starch, starch with gums, and gum-based). Rheological properties of thickened liquids can change over time. Stability is a fundamental characteristic of thickened liquids since changes in viscosity over time can affect the safety of swallowing those liquids [16]. The possibility of measuring the stability of thickeners over time is of great relevance because in many social and health centers, preparations are made in the morning to last the entire day, and many hours may pass between preparation and intake. Therefore, it is considered of clinical interest to study how, over time, these thickeners behave with water. For example, starch-based thickeners with water do not appear to be stable [4,16].

The aims of this work were: (1) to compare the rheological properties of four commercial thickeners dissolved in water under standard conditions, at different compositions and concentrations; (2) to formulate some equations that allow the prediction of viscosity reached according to the concentration, without the need for a rheometer, with adjustment to the Ostwald De-Waele power law; and (3) to analyze the evolution of the rheological properties over time, after preparation.

## 2. Material and Methods

### 2.1. Thickeners

We used four types of commercial thickeners: Resource ThickenUp^®^ (REA) and Resource ThickenUp Clear ^®^ (RC) from Nestlé Health Science Center, Barcelona, Spain; Delical gelodiet^®^ (GELO) from Lactalis Nutricion Iberia, Santander, Spain; and Nutilis clear^®^ (NC) from Nutricia, Madrid, Spain. REA is a first-generation thickener, and modified starch is used; GELO is second generation, and starch with gums are used; NC and RC are two third-generation thickeners, which use different types of gums and do not contain starch. The main ingredients of each one of the thickeners are as follows: (1) REA: modified corn starch; (2) GELO: modified corn starch, sodium chloride, native chicory inulin, guar gum, and konjac gum; (3) NC: maltodextrins, guar gum, xanthan gum, potassium chloride, and sodium chloride; and (4) RC: maltodextrins, xanthan gum, sodium chloride, and potassium chloride.

### 2.2. Solvent

We dissolved the thickeners in commercial mineral water from the Sacalm spring in Sant Hilari (Fontvella^®^, Girona, Spain) with the following composition: calcium 43.2 mg/L, magnesium 11.5 mg/L, sodium 12.3 mg/L, and bicarbonates 167 mg/L. It also had a conductivity of 303 μS/m. The water temperature was always in a range between 22 °C and 25 °C. Although ordinary water is used in practice, commercial drinking water was used for the study to maintain a standardized composition.

### 2.3. Preparation of the Samples

Preparation of the samples was done according to the methodology already described [12]. Briefly, water and thickeners were mixed in a shaker specially designed to dissolve thickeners in liquids, with a height of 16 cm, a diameter of 22 cm, and a capacity of 400 mL. The samples were weighed on a Nahita Blue Series 5173 precision electronic scale and then shaken 15 times with an arc of approximately 50 cm, trying to reproduce the real conditions of preparation as accurately as possible. The concentrations used for the preparation of the samples in water were between the minimum recommended for a thickener with nectar texture (1.2%), to the maximum recommended for a thickener with pudding texture (9%). The manufacturer’s recommendations use somewhat inaccurate measurements (“ladles”), which imply a high degree of subjectivity. As well as this, we added both intermediate concentrations, and a very low concentration (0.5%), in order to obtain as accurate a concentration/viscosity curve as possible. Starch-based thickeners could not be measured at concentrations below 2.4% because the resulting viscosity was too low. Following this method, the viscosity of the four thickeners could be compared with the same concentration values. All the concentrations tested are shown in Table 1.

The manufacturer’s recommendations on product labels are in bold for obtaining the different textures of nectar, honey, and pudding, according to the levels of The National Dysphagia Diet Task Force (NDD) [1]. For the nectar texture, the concentrations recommended by the manufacturers for NC, RC, REA, and GELO were 1.5%, 1.2%, 4.5%, and 4.5%, respectively. For the honey texture they were 3%, 2.4%, 6.9%, and 5.75%, respectively. Finally, for the pudding texture it was 4.5%, 3.6%, 9%, and 6.9%, respectively.

All the formulations were prepared in 200 mL of water, produced in triplicate, and placed in 200 mL beakers after preparation. Following preparation, the samples settled for 10 min before their subsequent analysis. We repeated the measurements at 6 h, aiming to evaluate the evolution of viscosity over time.

### 2.4. Rheological Analysis

A stress-controlled rheometer (MCR 301, Anton Paar Physica, Graz, Austria) equipped with a CC17 coaxial cylinder geometry was used. The dimensions of the cylindrical diameter were as follows: internal cylinder diameter: 16.7 mm; internal cylinder length: 25 mm; and external cylinder diameter: 18.1 mm.

Flow curves were drawn with a range between 0.01 and 200 s^−1^, plotting the shear stress and viscosity against the shear rate. Viscosity was determined at a shear rate of 50 s^−1^ obtained from each concentration’s flow curve. The temperature of the samples varied between 22 °C and 25 °C, in an attempt to emulate the most common temperature conditions during home and clinical consumption [17]. Three specimens of each sample were measured for each determination. Rheological measurements were determined after 10 min of sample preparation [12].

We examined the average viscosity of the different thickeners at 50 s^−1^ comparing those based on gums (NC vs. RC) with those based on starch (REA vs. GELO), since the differences between gum-based thickeners versus starch-based thickeners were obvious.

In the experiments on the evolution of viscosity over time, measurements were taken at 6 h after sample preparation, in addition to the basal measurement.

### 2.5. Ethical Aspects

Evaluation by the ethics committee was not required, because no people or animals were involved.

### 2.6. Statistical Analysis

The statistical software IBM^®^ SPSS^®^ Statistics 26 (IBM Corporation. Armonk, NY, United States of America) was used. Quantitative variables were described by means (standard deviations), and qualitative variables by means of proportions. It was considered that no variable followed the normal distribution since the number of determinations was 3. Differences were considered significant with a *p* < 0.05.

Means were compared using the non-parametric Kruskal Wallis and Mann-Whitney U tests. For the comparison of repeated means, the Wilcoxon sign rank test was used as a non-parametric test. The equations for calculating the theoretical viscosity from the analyzed data were obtained using the simple and exponential regression line.

The flow curve data were fitted to the rheological model of the power law (Ostwald-de Waele; *Shear Stress* = *K* × (*Shear Rate*)*n*), where shear stress is the shear stress, *K* is the consistency index, and *n* is the flow behavior index. The adjustment was made using GraphPad PRISM^®^ (GraphPad Software, Inc., San Diego, CA, USA).

## 3. Results

### 3.1. Viscosity of Thickeners at a Shear Rate of 50 s^−1^

The average viscosity achieved by each type of thickener at a shear rate of 50 s^−1^ with different concentrations is shown in Table 2. These data were obtained from the flow curves.

With the same concentration of thickener, each brand presented a different viscosity at 50 s^−1^. Among the gum-based thickeners, NC reached a higher viscosity than RC in all concentrations (*p* < 0.05), except at the concentration of 0.5%. Of the commercial thickeners containing primarily starch (GELO and REA), GELO showed lower viscosity than REA at lower than the recommended doses (*p* < 0.05), but when used at recommended doses its viscosity was significantly higher than REA with equal concentrations (*p* < 0.05). The dispersion values reflected in the standard deviation of the means obtained in the three repeated samples of the same concentration were higher, with elevated concentrations (6.9%) for the gum-based thickeners (REA and GELO).

Regarding the viscosities reached at 50 s^−1^ with different concentrations of thickeners (Figure 1), the relationship between the behavior of thickeners with gums (NC and RC) and concentration was observed to be linear, meaning that by increasing the amount of thickener, the viscosity of the sample increased proportionally. However, viscosity did not reach pudding texture, despite using high concentrations (9%). In contrast, the viscosity of the starch-based thickeners (REA and GELO) grew exponentially with the concentration (Figure 1).

### 3.2. Flow Curves and Adjustments to the Power Law of Thickeners Dissolved in Water

The data from the flow curves were fitted to the power law model (Ostwald–de Waele), the formula of which is:*Shear Stress* = *K* × (*Shear Rate*)*n*
where *Shear Stress* is equal to the shear stress, *K* is equal to the consistency index. *Shear Rate* = *shear rate* and *n* is the flow behavior index.

As can be seen in Figure 2, Figure 3, Figure 4 and Figure 5, the thickeners studied presented different rheological behaviors. NC and RC (based on gums) increased shear gradually with increasing concentrations. In contrast, the flow curves of the modified starch-based thickeners (GELO and REA) hardly increased the shear stress with increasing concentrations, except for higher concentrations (6.9% in the case of REA and 5.75% in the case of GELO), where the increase was significant (*p* < 0.01 and *p* < 0.0001, respectively).

The experimental results for shear stress and shear rate fitted well with the simple power law model, with high coefficients of determination (*R*^2^ > 0.9).

As can be seen in Table 3, the values obtained for *n* (flow behavior index) of all the thickeners were lower than <1, so it can be corroborated, according to this model, that the behavior of the four thickeners under study is pseudoplastic (as suggested in other sections). It should be noted that *n* decreased with greater thickener concentration, except in the case of the modified starch-based thickener (REA), where *n* increased with greater concentration. This may be due to the complex rheological behavior of starch, which exhibits dilating behavior at high concentrations. Among the gum-based thickeners (NC and RC), the highest *K* values (consistency index) obtained were those corresponding to the thickener based on xanthan gum, maltodextrins, and guar gum (NC). This would explain the higher viscosity values obtained in said thickener compared to RC. Among the starch-based thickeners (REA and GELO), the highest *K* values were for the modified starch-based thickener and gums (GELO).

### 3.3. Evolution of Viscosity over Time

Table 4 shows the viscosities obtained with the three thickeners in different textures and at three different times after their preparation. Six hours after their preparation, the gum-based thickeners (NC and RC) fluctuated between a decrease of 2.4% in their viscosity (RC in nectar texture) and an increase of 6.5% (NC in nectar texture), while the starch-based thickeners increased their viscosity up to 43.1%. This increase in viscosity of gum-based thickeners in honey or nectar textures was statistically significant (Table 4), although without clinical importance because it remains within the expected texture range.

## 4. Discussion

In this study, we compared the rheological properties of four thickeners commonly used in clinical practice, and their stability over time. The rheological properties of thickeners depend on their concentration, swallowing strength (shear rate), and time since preparation. The understanding of these characteristics can help to make correct use of thickeners and facilitate an individualized prescription. At a shear rate that can be the physiological speed of swallowing, the viscosity of gum-based thickeners (NC and RC) had a linear relationship with concentration. However, the viscosity of starch-based thickeners (REA and GELO) grew exponentially with increasing concentration. Describing this situation is very important, because when using starch-based thickeners in preparations, we have to be very precise with the quantity so as not to obtain undesirable viscosities.

Between the two gum-based thickeners analyzed, NC showed a higher thickening power at all concentrations less than 0.5% (Figure 1). This fact may be due to a higher thickening power of guar gum compared with xanthan gum, as shown by the studies by Park et al. and Seo et al. [18,19]. In these studies, guar gum-based thickeners were compared with xanthan gum-based thickeners, and thickeners with guar gum in their composition obtained higher viscosity values. Gum-based thickeners reached viscosities close to 800 mPa·s with less quantity than starch-based thickeners. Only at viscosities higher than 1000 mPa·s do starch-based thickeners obtain much higher viscosities than gum-based thickeners.

Starch-based thickeners (REA and GELO) obtained low viscosities at concentrations below 4%, but high viscosities when they reached concentrations above 6%. It is at these concentrations when the highest dispersion values are also appreciated. These results coincide with those of Garcia Gonzalez et al. [20], where the degree of dispersion of viscosity values obtained in measurements of water thickened with starch-based thickener from concentrations 6% and above, suggesting some variability of starch-based thickeners at high doses, probably due to worse solubility. Concentrations lower than 2.4% were not analyzed due to poor effect on dissolution.

Sopade et al. [21] described almost identical viscosity values to those obtained in our work for the starch-based thickener in water with a shear rate of 50 s^−1^. Furthermore, when analyzing a thickener based on modified starch with a mixture of gums similar to the one we used (GELO), Sopade obtained results very similar to our study with regard to low and medium concentrations (163.9 mPa·s at 3.6% and 426.1 mPa·s at 4.4%), but different in higher concentrations (5185.4 mPa·s at 6.9%). These differences between the results of studies may be due to multiple variables such as temperature, pH, rheology equipment with which the results were determined, or the ionic charge with which the thickener solution was carried out [22].

When we analyzed the viscosities reached at 50 s^−1^ with different concentrations of thickeners, the starch-based thickeners (REA and GELO) showed an exponential behavior with the concentration, so that with low doses of thickener, lower viscosities were achieved than with the gum-based thickeners (NC and RC). However, with high doses of thickener, the viscosity increased in a non-linear way, reaching much more elevated viscosities than those expected according to the manufacturer’s recommendations. Similar results were obtained by García González et al. [20], who described how the viscosity measured at 50 s^−1^ in a modified starch-based thickener increased exponentially with concentrations of 3%. This fact can lead to incorrect handling of the product, since small differences in thickener concentration can result in disparate viscosity values.

As mentioned above, to obtain water viscosities corresponding to nectar and honey textures (around 1000 mPa·s), the most efficient option would be to use gum-based thickeners, since less quantity of product is needed. However, certain factors must be taken into account (taste, smell, turbidity, stability, etc.) which may influence its behavior. To obtain higher viscosities of the pudding type (>1750 mPa·s) the best option could be to use thickeners based on modified starch. Conversely, according to Clavé and García [23], nectar and honey are the most commonly prescribed textures, with the pudding texture being the least used (6%). Pudding-type textures are less tolerated and unpalatable to patients, which increases the risk of dehydration and the level of satiety [23,24,25]. It has even been suggested that xanthan gum-based thickeners are safer than modified starch-based thickeners, since starch-based thickened fluids leave more residue in the mouth than xanthan gum, indicating better tolerability of thickened gum-based fluids by patients [15]. In addition, a higher viscosity will require a greater shear stress for correct swallowing, whereas the elderly have less strength. Viscosities greater than 800–1000 mP.as do not report greater benefits in the safety and efficiency of swallowing, according to some recent articles [13,26]. These viscosities are achievable by gum-based thickeners.

The Ostwald–de Waele power law model shows that the rheological behavior of the thickeners under study is pseudoelastic, and the value *n* < 1 with a high correlation (R^2^ > 0.9). This behavior indicates that the viscosity decreases when an external shear force is applied to it, such as when the fluid enters the mouth and is propelled towards the pharynx [27]. This fact is relevant, since almost all quantitative texture classifications suggest measurements at a shear rate of 50 s^−1^. Consistently measuring viscosities at this shear rate can be a mistake, since the speeds vary greatly depending on the patient and the phase of swallowing, even reaching values of 900s^−1^ [25]. The pseudoplastic behavior mentioned above has been described for most thickened liquids [28,29], both in thickeners based on xanthan gum and gum arabic [30], and in thickeners with starch in their composition [21]. The behavior of the modified starch-based thickener (REA) was analyzed by Sopade et al. [21], where they found the behavior of the starch at some points was *n* > 1, by applying the Generalized Herschel-Bulkley model. This value would indicate dilatant fluid behavior (an increase in viscosity with shear rate). The characteristic that thickened fluids exhibit a non-Newtonian behavior is of great importance from the clinical point of view, to choose the method to measure said viscosity. The NDD recommends the use of a rheometer, but the International Dysphagia Diet Standardization Initiative (IDDSI) proposes a simple method by assessing the rate of fall of the thickened fluid through a syringe, where the shear stress is much lower [31]. Evaluating viscosity by rheometer has the advantage that it provides the shear factor, as occurs in the swallowing process. Other methods observe the viscosity of the fluid according to the rate of fall due to the force of gravity. In view of our results, it can be concluded that to obtain viscosities corresponding to nectar and honey textures, gum-based thickeners need lower concentration than starch-based, in order to reach the same texture. Therefore, gum-based thickeners could be more efficient when used in clinical practice. On the other hand, in cases where pudding texture is needed, starch-based thickeners would be more efficient because gum-based thickeners do not reach the high viscosities required.

The most common textures in clinical practice are nectar and honey [23]. Variations over time between gum-based thickeners and exclusively modified starch at these two textures are also of interest. Our study has demonstrated that the viscosity of water solutions with starch-based thickeners have a complex behavior, because it increases in such a way that it is no longer reliable and can limit its use. In contrast, the viscosity of water with gum-based thickeners remains constant for six hours, so healthcare workers can use it with confidence throughout their work shifts. The usefulness of the results obtained in terms of the evolution of thickeners over time is important for daily clinical practice. In reality, it is not uncommon for the thickened water to remain on the patient’s bedside table for use throughout the day, because patients with dysphagia ingest water with difficulty and effort. In addition, the workload of nursing homes sometimes means that thickened water is prepared, together with medication, many hours in advance. It is therefore extremely important to present a study investigating the rheological behavior of the consistencies that can be achieved with the products offered on the market, which will provide complete and understandable information to healthcare professionals for the recommendation of a diet adapted and tailored to each patient by taking into account textures, viscosities, and volumes.

The viscosity variations observed over time with starch-based thickeners have already been described by other authors, with occasionally controversial results. In our study, determinations were made at 6 h to have a better understanding of what is happening with the viscosity of the thickened water throughout the day. Other works [27] have evaluated the variation of viscosity in the first 10 and 30 min after being mixed, because they considered it to be a reasonable time for the patient to take before using the preparation. These other studies used three starch-based thickeners and two gum-based thickeners. Garcia et al. [27,32] observed that gum-based thickeners with water, both with the nectar and honey consistency, did not vary significantly (*p* > 0.05) over time, while they noticed a significant (*p* < 0.05) change with starch-based thickeners, some doubling their viscosity within 30 min of preparation. However, Dear and Joyce [16], when studying the viscosity evolution of three starch-based thickeners with tap water every 30 min, over a 16 h period, observed a viscosity decrease in the first 30 min of 33% with a concentration of 4 g in 200 mL, and of 18% with 6 g in 200 mL. Subsequently, the viscosity progressively increased, reaching 42 and 50% for the respective concentrations. The thickener with maltodextrins did not present this decrease in initial viscosity, but subsequently increased. These variations are important because they can change the range of the expected viscosity.

Furthermore, O’Leary et al. [28] studied the variation of viscosity every 15 min over a 3 h period after the preparation of several starch-based thickeners (one of them REA) and another thickener with starch, gums, and maltodextrins. Results among the starch-based thickeners were highly variable, with increases of 15 to 37% and even decreases of 34%. Sopade et al. [21] compared six commercial thickeners: two thickeners with guar gum, two with xanthan gum, and two with modified starch (one of which was similar to REA), without finding significant (*p* > 0.05) variations between any of them at 24 h. It was not established whether the observed variations in viscosity and the differences between the brands of thickeners were due to the constituent ingredients, to a greater dissolution of the powder over time, to the evaporative concentration of the fluid, or changes in the nature of the chemical bonds.

Finally, Alves et al. [29] studied the viscosity of a thickener based on maltodextrin, xanthan gum, and potassium chloride (similar to RC). For the determination of the viscosity, they did not use a rheometer but instead the method developed by the International Dysphagia Diet Standardization Initiative based on syringes. Viscosity was determined after the preparation, then every hour for the first 12 h, and then daily at 08:00 for the following four days. A small increase in consistency was observed 10 h after the preparation, and then a bit of a decrease at 24 h and 48 h. However, these results have their limitations because of the technique used for their determination.

This study provides as a novelty the rheological comparison with a wide range of concentrations and the management characteristics of four different types of thickeners, representative of the thickeners for clinical use which the market offers. Most of the works previously published are limited to one or two thickeners, and to very specific concentrations. This has allowed us to describe for the first time that gum-based thickeners have a linear relationship between their concentration and viscosity, while starch-based thickeners have an exponential one. Finally, we present the regression curves for the different water-based thickeners that predict the rheological results of this work, thus eliminating the necessity of performing rheological determinations in the laboratory any longer.

However, there are some limitations. It was not possible to determine whether the different rheological characteristics are associated with greater safety during swallowing. Studies by videofluoroscopy would be necessary to evaluate the risk of aspiration with the different viscosities. In addition, it would be interesting to carry out measurements at speeds other than 50 s^−1^ in future studies. Another limitation of the study was the lack of evaluation of the early changes in viscosity, as we only performed determinations at the beginning and 6 h. Nevertheless, we considered it more important, from a practical point of view, to determine the preparation stability during a regular healthcare worker shift. 

## 5. Conclusions

The viscosity of gum-based thickeners is quite predictable, because there is a linear relationship between the viscosity achieved and the concentration of the thickener in water. The higher the concentration, the higher the viscosity achieved. This makes gum-based thickener much easier to handle, although it is very difficult to achieve pudding-like texture. In addition, there are rheological differences between thickeners based on gums with different compositions. Conversely, the viscosity obtained with starch-based thickeners is not so predictable, since there is an exponential relationship between viscosity and concentration. At low and medium concentrations, the viscosity of starch-based thickeners in water is very low, but at high concentrations, the viscosity rises to very thick pudding-like textures. This can complicate handling in daily clinical practice, especially if concentrations in the preparation are not well calculated.

To achieve viscosities in water corresponding to the textures of nectar and honey, the best option could be to use thickeners based on gums, and not on starch, since less product is needed to achieve the desired results. Starch-based thickeners are best to achieve very thick textures. However, they are not stable over time (their viscosity increases), and they add flavor, color to the water, unpalatability, and pharyngeal residue.

The changes observed in viscosity over time have clinical implications, since thickened liquids are often not immediately consumed after mixing. In this sense, gum-based thickeners are much more stable than starch-based thickeners, the viscosity of which increases over time. In order to be sure that the modified texture viscosity is as desired for clinical use, it is recommended that thickened liquids are consumed within a short period of being mixed, otherwise they should be withdrawn and prepared again. The formulas presented allow estimation of the desired concentration or viscosity for these types of thickeners, which can be very useful for handling and individualizing the prescription.

Further studies are needed to verify whether the rheological characteristics of thickeners influence safety, adherence, or patient preference for different thickeners according to the type or intensity of their dysphagia.

## Figures and Tables

**Figure 1 nutrients-14-03455-f001:**
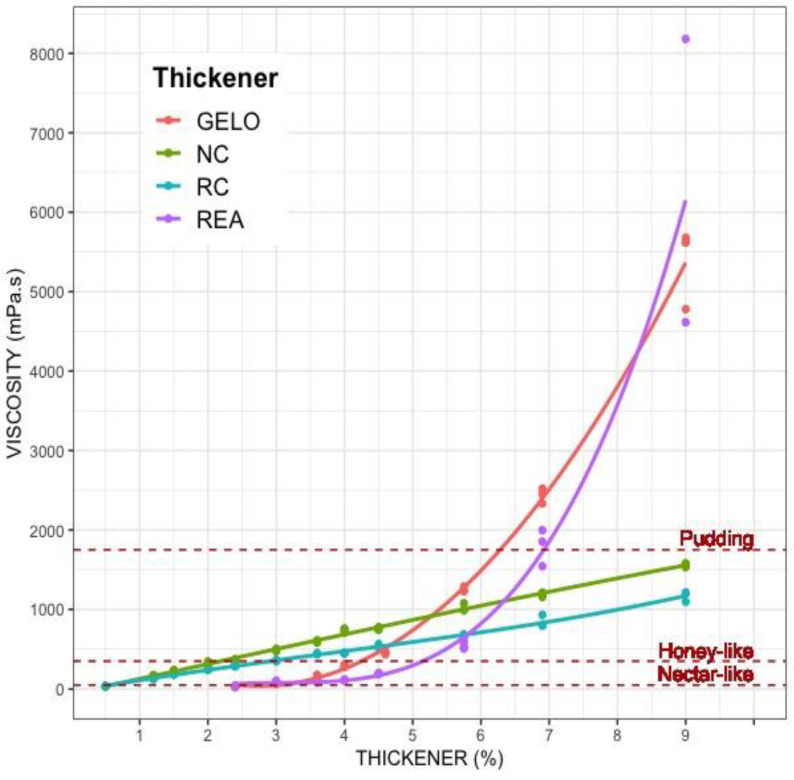
Viscosity of the four thickeners (measured at 50 s^−1^) as a function of concentration. The viscosities determined at each concentration are shown with dots linked by their regression curve. Starch-based thickener (GELO in orange, REA in purple.), Gum-based thickener (NC in green, RC in blue). Regression curves were obtained for the different thickeners with water as a function of concentration. With these curves, it is possible to calculate the amount of thickener necessary to obtain a given viscosity at 50 s^−1^: NC: *V* = (180 × X) − 43.08 with a *R*^2^ of 0.995. RC: *V* = (128 × *X*) − 25.44 with a *R*^2^ of 0.989. REA: *V* = 2920 − (1620 × X) + (219 × X^2^) with a *R*^2^ of 0.920. GELO: *V* = 1830 − (1060 × X) + (167 × X^2^) with a *R*^2^ of 0.998. *X* represents the concentration of thickener (in %.).

**Figure 2 nutrients-14-03455-f002:**
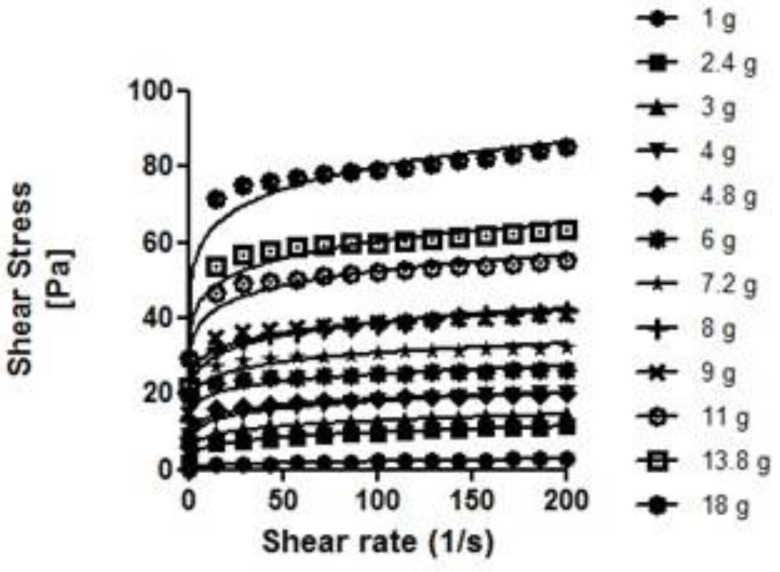
Flow curves at different concentrations of NC in water. Solid line represents the Ostwald de Waele rheological model.

**Figure 3 nutrients-14-03455-f003:**
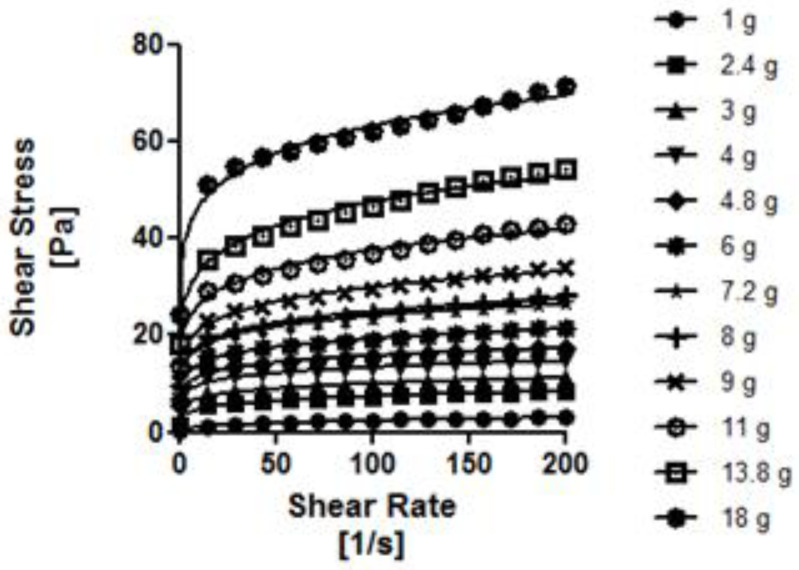
Flow curves at different RC concentrations in water. Solid line represents the Ostwald de Waele rheological model.

**Figure 4 nutrients-14-03455-f004:**
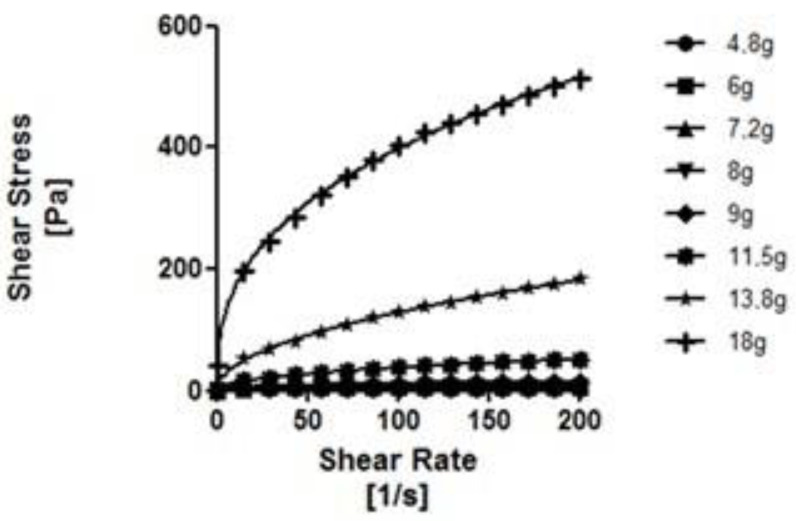
Flow curves at different concentrations of REA in water. Solid line represents the Ostwald rheological model de Waele.

**Figure 5 nutrients-14-03455-f005:**
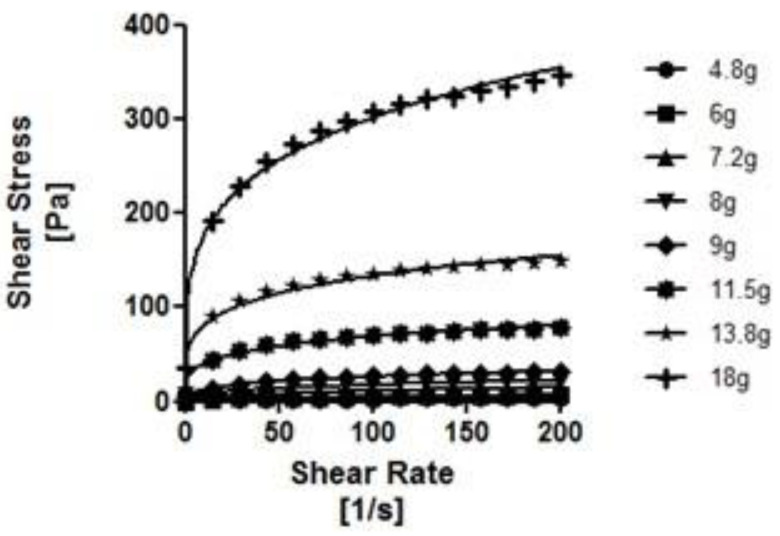
Flow curves at different concentrations of GELO in water. Solid line represents the Ostwald rheological model de Waele.

**Table 1 nutrients-14-03455-t001:** Concentrations of thickeners made with water.

Samples	Concentration of the Thickener (%)											
NC	0.5%	1.2%	1.5%	2%	2.4%	3%	3.6%	4%	4.5%	5.75%	6.9%	9%
RC	0.5%	1.2%	1.5%	2%	2.4%	3%	3.6%	4%	4.5%	5.75%	6.9%	9%
REA					2.4%	3%	3.6%	4%	4.5%	5.75%	6.9%	9%
GELO					2.4%	3%	3.6%	4%	4.5%	5.75%	6.9%	9%

NC: Nutilis clear^®^; RC: Resource ThickenUp Clear ^®^; REA: Resource ThickenUp^®^; GELO: Delical gelodiet^®^.

**Table 2 nutrients-14-03455-t002:** Average viscosity of the four thickeners at a 50 s^−1^ shear rate and comparisons between gum-based thickeners (NC vs. RC) and starch-based thickeners (REA vs. GELO).

Concentration (%)	NC Viscosity Mean (SD) mPa·s	RC Viscosity Mean (SD) mPa·s	Differences NC vs. RC	REA Viscosity Mean (SD) mPa·s	GELO Viscosity Mean (SD) mPa·s	Differences REA vs. GELO
**0.5%**	34.2 (2.4)	37.9 (1.6)	* **3.70** *	-	-	
**1.2%**	169.6 (8.5)	131.7 (7.1)	* **37.93 ^b^** *	-	-	
**1.5%**	233.9 (4.1)	183.9 (1)	* **49.96 ^d^** *	-	-	
**2%**	327.1 (25.1)	247.3 (5.3)	* **79.85 ^a^** *	-	-	
**2.4%**	352.3 (23.5)	292 (3.4)	* **60.28 ^a^** *	40.2 (4.2)	29.3 (5.8)	* **10.88 ^a^** *
**3%**	491.3 (15.5)	356.6 (6.2)	* **134.68 ^d^** *	97.3 (8.4)	68.6 (3.6)	* **28.68 ^b^** *
**3.6%**	599 (14.4)	441.6 (8.4)	* **157.33 ^d^** *	106.3 (15.1)	169.3 (10.2)	* **62.98 ^b^** *
**4%**	736.2 (27.1)	450.1 (6.16)	* **286.15 ^d^** *	113.9 (8.5)	291.7 (15.9)	* **177.73 ^d^** *
**4.5%**	755.5 (18.7)	539.9 (22.6)	* **215.56 ^d^** *	194.7 (6)	448.9 (16)	* **254.20 ^d^** *
**5.75%**	1020 (49.2)	669.6 (17)	* **350.46 ^d^** *	542.9 (4.49)	1250.6 (33.2)	* **707.71 ^d^** *
**6.9%**	1184.4 (27.5)	843.3 (75.9)	* **341.18 ^b^** *	1797.6 (230.8)	2434 (94.3)	* **636.33** *
**9%**	1559.5 (24.2)	1168.4 (62.6)	* **391.05 ^c^** *	6139 (1838)	5358.8 (502.1)	* **780.83** *

Note: Viscosity was presented in means (standard deviations). In bold and italic: Average differences. Significance level: *^a:^ p* < 0.05, *^b:^ p* < 0.01, *^c:^ p* < 0.001, and *^d:^ p* < 0.0001.

**Table 3 nutrients-14-03455-t003:** Corresponding parameters of the fit to the Ostwald–de Waele model.

Thickener	Concentration (%)	Parameters		
		*K (SD)*	*N (SD)*	*R* ^2^
NC	0.0%	0.39 (0.02)	0.36 (0.01)	0.998
	1.20%	3.59 (0.15)	0.21 (0.009)	0.998
	15.0%	5.50 (0.30)	0.17 (0.01)	0.994
	2%	8.51 (0.38)	0.16 (0.009)	0.995
	2.40%	10.14 (1.02)	0.13 (0.02)	0.962
	3%	14.85 (1.94)	0.11 (0.02)	0.916
	3.60%	18.10 (2.19)	0.11 (0.02)	0.928
	4%	21.16 (1.64)	0.13 (0.01)	0.977
	4.50%	23.64 (2.33)	0.10 (0.02)	0.941
	5.75%	31.61 (3.39)	0.10 (0.02)	0.934
	6.90%	36.40 (4.29)	0.10 (0.02)	0.922
	9%	48.06 (5.30)	0.11 (0.02)	0.933
RC	0.50%	0.47 (0.01)	0.35 (0.007)	0.999
	1.20%	3.09 (0.25)	0.18 (0.01)	0.989
	1.50%	4,86 (0.43)	0.15 (0.01)	0.980
	2%	7.18 (0.44)	0.13 (0.01)	0.986
	2.40%	8.46 (0.57)	0.13 (0.01)	0.983
	3%	10.21 (0.52	0.13 (0.01)	0.991
	3.60%	12.69 (0.68)	0.13 (0.01)	0.990
	4%	12.98 (0.64)	0.14 (0.01)	0.991
	4.50%	14.83 (0.69)	0.15 (0.01)	0.993
	5.75%	18.44 (1.22)	0.15 (0.01)	0.987
	6.90%	23.03 (1.64)	0.15 (0.01)	0.986
	9%	34.11 (1.8)	0.13 (0.01)	0.989
GELO	2.40%	0.11 (0.01)	0.64 (0.02)	0.998
	3%	0.46 (0.06)	0.50 (0.02)	0.996
	3.60%	1.47 (0.02)	0.44 (0.03)	0.993
	4%	3.67 (0.55)	0.34 (0.03)	0.991
	4.50%	6.40 (1.20)	0.30 (0.04)	0.983
	5.75%	23.68 (4.38)	0.23 (0.03)	0.970
	6.90%	49.85 (7.97)	0.21 (0.03)	0.973
	9%	103.2 (12.25)	0.23 (0.02)	0.987
REA	2.40%	0.658 (0.12)	0.26 (0.04)	0.998
	3%	1.646 (0.24)	0.25 (0.03)	0.996
	3.60%	1.798 (0.28)	0.25 (0.03)	0.993
	4%	1.873 (0.32)	0.26 (0.03)	0.991
	4.50%	2.267 (0.35)	0.36 (0.03)	0.983
	5.75%	4.776 (0.45)	0.45 (0.01)	0.970
	6.90%	13.17 (1.66)	0.49 (0.02)	0.973
	9%	72.57 (3.82)	0.37 (0.01)	0.987

**Table 4 nutrients-14-03455-t004:** Representation of average viscosity in mPa·s as a function of time (10 min, 6 h) after preparation, measured at a shear rate of 50 s^−1^ and obtained with different textures.

Thickeners Concentration (Texture)	10 min from Preparation	6 h from Preparation	Differences 10 m vs. 6 h Mean (SD) and (Variation Percentage)
RC 2% (nectar)	247.3 (5.3)	241.3 (3.5)	−5.9 (3) (−2.4%) *^a^*
NC 2% (nectar)	327.1 (25.1)	348.7 (29)	+21.5 (4.9) (+6.5%) *^a^*
REA 4.5% (nectar)	194.7 (6)	265.2 (10.7)	+70.4 (0.6) (+36.2%) *^b^*
RC 4% (honey)	450.1 (6.1)	436.3 (7.1)	−13.8 (1.9) (−3.1%) *^b^*
NC 4% (honey)	736.2 (27.1)	768.4 (23.5)	+32.1 (3.6) (+4.3%) *^b^*
REA 6.5% (honey)	1697.6 (134)	2463 (135.1)	+732 (269.6) (+43.1%) *^d^*

Note: Viscosity presented in means (standard deviations), and in brackets are the differences expressed in percentage. Significance level: *^a:^ p* < 0.05, *^b:^ p* < 0.01 and *^d:^ p* < 0.0001.

## Data Availability

Data supporting reported results are available on request from the corresponding author.

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
