# Peer review of "Rheological Properties and Stability of Thickeners for Clinical Use"

_nutrients, 2022, doi:10.3390/nu14173455_

Round 1

Reviewer 1 Report

Manuscript ID: Nutrients_ 1861340

In the article entitled: “Rheological behavior in different commercial thickeners for clinical use according to their concentration, composition and stability over time, with a predictive model” was study the rheological behavior of various commercial thickeners for patients with dysphagia. The article is interesting.

This article has been written in a compact manner. From the methodological point of view, the employed measurement techniques are appropriate to the adopted objective of the research work. The results obtained may have practical application. Research findings may improve the quality of life of people suffering from dysphagia.

Title

The title and the aim of the study are clearly constructed.

Abstract

The abstract includes the aim of the study, methods used in the experiment and contain the principal results and conclusions.

Introduction

The introduction describes the matter of the experiment and states the problem being investigated. The cited literature refers to the subject of the analyzed problem.

Methods

The data is well collected. The methods as far as possible described in detail (below I have questions). The sampling is appropriate and adequately described. Statistical analysis of measurement results has been used.

2.2. Solvent

Why was mineral water used and not distilled water? In practice, ordinary water is used.

2.3. Preparation of the samples

Thickeners were dissolved in the cold? Starch-based thickener also? Was the suspension mixed during preparation? There is a literature reference, but it would be advisable to describe.

Results and Discussion

Authors, generally correctly interpreted and described the significance of the results for the research. However, I have a comment:

 3.2. Flow curves and adjustments to the power law of thickeners dissolved in water

the paragraph 203: “This may be due to the complex rheological behavior of starch.” Maybe some attempt to explain? Maybe one sentence?

3.3. Evolution of viscosity over time

It is a pity that the research was done in the initial and final time. The first hours are important in the kinetics of structuring.

the paragraph 313-316: The authors refer to the stability of the studied systems but was stability studied? How? What method?

Conclusion

The authors correctly indicate, how the results are related to the studies.

References

Generaly the references are accurate.

Language

The article is correctly written. English language and style are minor spell check required.

Author Response

We wanted to thank the reviewers for their hard and selfless work, even in the summer and with these heat waves.

Their work is essential to improve the quality of the articles to be published.

Below we collect the comments of the reviewers and our answers marked in blue

Reviewer 1

Manuscript ID: Nutrients_ 1861340

In the article entitled: “Rheological behavior in different commercial thickeners for clinical use according to their concentration, composition and stability over time, with a predictive model” was study the rheological behavior of various commercial thickeners for patients with dysphagia. The article is interesting.

This article has been written in a compact manner. From the methodological point of view, the employed measurement techniques are appropriate to the adopted objective of the research work. The results obtained may have practical application. Research findings may improve the quality of life of people suffering from dysphagia.

Title

The title and the aim of the study are clearly constructed.

Abstract

The abstract includes the aim of the study, methods used in the experiment and contain the principal results and conclusions.

Introduction

The introduction describes the matter of the experiment and states the problem being investigated. The cited literature refers to the subject of the analyzed problem.

Methods

The data is well collected. The methods as far as possible described in detail (below I have questions). The sampling is appropriate and adequately described. Statistical analysis of measurement results has been used.

2.2. Solvent

Why was mineral water used and not distilled water? In practice, ordinary water is used.

Thank you very much for your comments. They enrich and clarify the article.

As the reviewer indicates, ordinary water is usually used in practice, but in our study, we always used the same mineral water from a certain commercial house to standardize the results. We consider that ordinary water could change from one day to the next in pH or chlorination. On the other hand, we do not use distilled water because we thought it better to use potable water.

In line 105 to 107 of the article, we introduce the following sentence:

“Although ordinary water is used in practice, commercial drinking water was used for the study to maintain a standardized composition.”

2.3. Preparation of the samples

Thickeners were dissolved in the cold? Starch-based thickener also? Was the suspension mixed during preparation? There is a literature reference, but it would be advisable to describe.

The reviewer is right. Although the methodology in the preparation of the samples is described in a previous article of ours, given its importance, we repeat it in the article on line 110-115:.

“Preparation of the samples according to the methodology already described [12]. Briefly, water and thickeners were mixed in a shaker specially designed to dissolve thick-eners in liquids, with a height of 16 cm, a diameter of 22 cm and a capacity of 400 ml. The samples were weighed on a Nahita Blue Series 5173 precision electronic scale and then shaken 15 times with an arc of approximately 50 cm, trying to reproduce the real condi-tions of preparation  as accurately as possible”

We also add in the text the temperature at which the determinations were made:

The water temperature was always in a range between 22ºC and 25º C (line 104-105) .

Results and Discussion

Authors, generally correctly interpreted and described the significance of the results for the researchHowever, I have a comment:

 3.2. Flow curves and adjustments to the power law of thickeners dissolved in water

the paragraph 203: “This may be due to the complex rheological behavior of starch.” Maybe some attempt to explain? Maybe one sentence?

Thank you very much for the objection, we have added in line 237-238 the following explanatory sentence:  “This may be due to the complex rheological behavior of starch, which exhibits dilating behavior at high concentrations”

3.3. Evolution of viscosity over time

It is a pity that the research was done in the initial and final time. The first hours are important in the kinetics of structuring.

The reviewer is right. We add this consideration in the study limitations part in line 418-422

“Another limitation of the study was the lack of evaluation of the early changes in viscosity, as we only performed determinations at the beginning and 6 hours. Nevertheless, we considered more important, from a practical point of view, to determine the preparation stability during a regular healthcare worker shift”.

the paragraph 313-316The authors refer to the stability of the studied systems but was stability studied? How? What method?

The reviewer is correct that the expression “complex behavior and are unstable over time” is confused.

In paragraph 351-356, what we want to say is that 6 hours after its preparation, the viscosity varied in a clinically very significant way. For this reason we consider that this preparation cannot be kept for so long without using it.

We have changed the paragraph as follows:

“Our study has demonstrated that the viscosity of water solutions with starch-based thickeners have a complex behavior, because it increases in such a way that it is no longer reliable and can limit its use. In contrast, the viscosity of water with gum-based thickeners remains constant for six hours, so healthcare workers can use it with confidence throughout their work shifts”

 Conclusion

The authors correctly indicate, how the results are related to the studies.

References

Generaly the references are accurate.

Language

The article is correctly written. English language and style are minor spell check required.

Reviewer 2 Report

The authors present useful rheological properties of different thickeners with fitting regression models. The subject fits the theme of the journal, and this type of work is needed in the literature given the importance of thickener viscosity issues related to medical uses. However, the manuscript needs major revision to improve clarity. The followings offer suggestions and comments for the authors to consider.

The title is too long. I suggest “Rheological properties and stability of thickeners for clinical use”. And include “predictive mode”, “centration” etc. in keywords.

Authors need to revise the manuscript thoroughly to eliminate grammatical errors. For example,

1.      Please use past tense to describe methodology and results.

2.      L34, add “having” before “problems”

3.      L101, Preparation of the samples followed the methods described in [12].

4.      L115, delete “were” before “produced”

5.      Consistently use “dash”. For example, L39, starch-based, gum based

Abbreviations must be defined at their first mention (e.g., L86-93, REA, GELO, NC…)

Similarly, in Table 1, please explain all symbols and abbreviations used.

Table 1, were these compositions measured by authors themselves? If not, please include data resources (literature, datasheet, website…). Also, authors need to explain the significance and relevance of these compositions to viscosity. Otherwise, Table 1 is not necessary.

L95, is mineral water a common solvent? Why not pure water?

Table 2, L113 Please indicate which manufacturers and the source of their recommendations.

Please include briefly describe “nectar”, “honey” and “pudding” texture.

Table 2, most journals suggest that authors should avoid using shading in table cells. Please double-check check Guide for Authors.

L121, please indicate the dimension of the cylinder chamber.

Table 3, please replace “[%] with “concentration (%)”.

Figure 1: The caption should comprise a brief title and a description of the illustration. Keep text in the illustrations themselves to a minimum but explain all symbols and abbreviations used. 

Author Response

We wanted to thank the reviewers for their hard and selfless work, even in the summer and with these heat waves.

Their work is essential to improve the quality of the articles to be published.

Below we collect the comments of the reviewers and our answers marked in blue

Reviewer 2

The authors present useful rheological properties of different thickeners with fitting regression models. The subject fits the theme of the journal, and this type of work is needed in the literature given the importance of thickener viscosity issues related to medical uses. However, the manuscript needs major revision to improve clarity. The followings offer suggestions and comments for the authors to consider.

The title is too long. I suggest “Rheological properties and stability of thickeners for clinical use”. And include “predictive mode”, “centration” etc. in keywords.

We agree with the reviewer that the title is too long and we had many discussions between the authors. We wanted to draw the reader's attention to the novel aspects of our article. The solution found by the reviewer seems like a good idea to us and we are going to change it.

Authors need to revise the manuscript thoroughly to eliminate grammatical errors. For example,

  1. Please use past tense to describe methodology and results.
  2. L34, add “having” before “problems”
  3. L101, Preparation of the samples followed the methods described in [12].
  4. L115, delete “were” before “produced”
  5. Consistently use “dash”. For example, L39, starch-based, gum based

Thank you for your intense review we have corrected all the comments that you have recommended to us

Abbreviations must be defined at their first mention (e.g., L86-93, REA, GELO, NC…)

The reviewer is right and we have included the meaning of these acronyms both at the beginning of their use and in all footnotes to tables and figures.

Similarly, in Table 1, please explain all symbols and abbreviations used.

Table 1, were these compositions measured by authors themselves? If not, please include data resources (literature, datasheet, website…). Also, authors need to explain the significance and relevance of these compositions to viscosity. Otherwise, Table 1 is not necessary.

The complete composition of each thickener was provided to us directly by each commercial laboratory, because only its partial composition appears on its web pages. That's why we thought it useful to add it to the article.

On the other hand, it is true that the nutritional composition of these thickeners is not relevant for this study, so it can be perfectly removed.

L95, is mineral water a common solvent? Why not pure water?

Thank you very much for your comments. They enrich and clarify the article.

As the reviewer indicates, ordinary water is usually used in practice, but in our study we always used the same mineral bottled water from a certain commercial house to standardize the results. We consider that ordinary water could change from one day to the next in pH or chlorination, and we used drinking to try to recreate as best as possible the conditions of use of the product . The selected water was one of of the most consumed in Spain

In line 105to 107 of the article, we introduce the following sentence:

“Although ordinary water is used in practice, commercial drinking water was used for the study to maintain a standardized composition.

Table 2, L113 Please indicate which manufacturers and the source of their recommendations.

We have added in line 86-89  the manufacturers of each product

Please include briefly describe “nectar”, “honey” and “pudding” texture,

Table 2, most journals suggest that authors should avoid using shading in table cells. Please double-check check Guide for Authors.

The reviewer is right that what we have done needs to be better explained.

In Table 2 we removed the colors and the phrase "The manufacturer's recommendations for obtaining the different textures are shown in blue for nectar, green for honey and purple for pudding".

In line 127 we have added the following sentence:

The manufacturer's recommendations on product labels in bold for obtaining the different textures of nectar, honey and pudding according to the levels of the The National Dysphagia Diet Task Force (NDD) [1]. For nectar texture, the concentrations recommended by the manufacturers for NC, RC, REA and GELO were 1.5%, 1.2%, 4.5% and 4.5%, respectively. For honey texture they were 3%, 2.4%, 6.9% and 5.75% respectively. Finally, for pudding texture it was 4.5%, 3.6%, 9% and 6.9%.

L121, please indicate the dimension of the cylinder chamber

Thank you very much for the clarification, so the reader has more information, we add the next sentence in line 139-141;

“using a CC17 coaxial cylinder geometry with the following dimensions: internal cylinder diameter; 16.66 mm, internal cylinder length; 25.046 mm and external cylinder diameter; 18.079 mm”

Table 3, please replace “[%] with “concentration (%)”.

We have changed what the reviewer indicates

Figure 1: The caption should comprise a brief title and a description of the illustration. Keep text in the illustrations themselves to a minimum but explain all symbols and abbreviations used. 

Thank you for your comment. In this way we can emphasize something that seems very relevant to us: the behavior of thickeners based on their composition. The reader can graphically observe how the starch-based thikeners present this exponential behavior and the gum-based thickeners this linear behavior. This is very important when when using it.

We have added the following sentence in the footer:

Viscosity of the four thickeners (measured at 50 s-1) as a function of concentration. The viscosities determined at each concentration are shown with dots linked by their regression curve. Starch-based thickener (GELO in orange and REA in purple.), Gum based thickener (NC in green, RC in blue) 

Round 2

Reviewer 2 Report

The revised version of the manuscript has been well revised and improved, so it can be considered for publication after a minor revision.

*Line 139, please change this sentence to "A stress-controlled rheometer (MCR 301, Anton Paar Physica, City?, Austria) equipped with a CC17 coaxial cylinder was used. The dimension of the cylindrical chamber was..."

*Please bear in mind that each time when you use "significant" or "significantly", the p-value needs to be included next to it. For example, Line 219, 250, 376, etc. Please double-check. 

* Line-140-141, three-digit after the decimal is not necessary. Suggest using one significant digit, for example, 16.7 mm not 16.66 mm, 18.1 mm not 18.079 mm.

Author Response

Thank you very much for your review. We believe, like you, that this improves the publication.

The revised version of the manuscript has been well revised and improved, so it can be considered for publication after a minor revision.

*Line 139, please change this sentence to "A stress-controlled rheometer (MCR 301, Anton Paar Physica, ?, Austria) equipped with a CC17 coaxial cylinder was used. The dimension of the cylindrical chamber was..."

Thank you for your indication. We have added the Austrian city of Graz

A stress-controlled rheometer (MCR 301, Anton Paar Physica, Graz, Austria) was employed using a CC17 coaxial cylinder geometry with the following dimensions: internal cylinder diameter; 16.66 mm, internal cylinder length; 25.046 mm and external cylinder diameter; 18.079 mm.

* Line-140-141, three-digit after the decimal is not necessary. Suggest using one significant digit, for example, 16.7 mm not 16.66 mm, 18.1 mm not 18.079 mm.

We change the sentence as follows:

 "A stress-controlled rheometer (MCR 301, Anton Paar Physica, Graz, Austria) equipped with a CC17 coaxial cylinder was used. The dimension of the cylindrical chamber was: internal cylinder diameter; 16.7 mm, internal cylinder length; 25.05 mm and external cylinder diameter; 18.1 mm”

*Please bear in mind that each time when you use "significant" or "significantly", the p-value needs to be included next to it. For example, Line 219, 250, 376, etc. Please double-check. 

The reviewer is right. We add the statistical significance in the text.

Line 219

By contrast, the flow curves of the modified starch-based thickeners (GELO and REA) hardly increased the shear stress with increasing concentrations, except for higher concentrations (6.9% in the case of REA and 5,75% in the case of GELO), where the increase was significant (p<0.01 and p<0.0001, respectively).

Line 250

We have changed the sentence by adding the statistical significance in table 4.

*In bold and italic: Average differences. *Significance level: a= p<0.05, b= p<0.01, c=p<0.001 and d= p<0.0001.

This increase in viscosity of gum-based thickeners in honey or nectar textures was statistically significant (table 4), although without clinical importance because it re-mains within the expected texture range.

Line 376

Garcia et al. [27,32] observed that gum-based thickeners with water, both with the nectar and honey consistency, did not vary significantly (p>0.05)over time, while they noticed a significant (p < 0.05) change with starch-based thickeners, some doubling their viscosity within 30 minutes of preparation.

Line 385

Dear and Joyce [16] do not specify the degree of significance in their publication, so we have chosen to remove the word significantly.

Line 392

Sopade et al [21] compared six commercial thickeners: two thickeners with guar gum, two with xanthan gum and two with modified starch (one of which was similar to REA), without finding significant (p>0.05) variations between any of them at 24 hours

line 282

Garcia Gonzalez et al [20] do not specify the degree of significance in their publication, so we have chosen to remove the word significantly.
